# Neutrino Decoherence in Simple Open Quantum Systems

Bin Xu[*]

*Institute for Fundamental Theory, Department of Physics,*
*University of Florida, Gainesville, FL 32611, USA*

## Abstract

Neutrinos lose coherence as they propagate, which leads to the fading away of oscillations. In this work, we model neutrino decoherence induced in open quantum systems from their interaction with the environment. We first present two different models in the quantum mechanical framework, in which the environment is modeled as forced harmonic oscillators with white noise interactions, or two-level systems with stochastic phase kicks. We then look at the decoherence process in the quantum field theoretic framework induced by elastic scatterings with environmental particles. The exponential decay is obtained as a common feature for all models, which shows the universality of the decoherence processes. We discuss connections to the GKSL master equation approach and give a clear physical meaning of the Lindblad operators. We demonstrate that the universality of exponential decay of coherence is based on the Born-Markov approximation. The models in this work are suitable to be extended to describe real physical processes that could be non-Markovian.

[*] Email: binxu@ufl.edu

## I. INTRODUCTION

Neutrino oscillations are a phenomenon well established theoretically as well as experimentally [1–3], caused by the mixing between the neutrino mass and flavor eigenstates:

$$|\nu_\alpha\rangle = \mathcal{U}^*_{\alpha i}|\nu_i\rangle,$$

where $\alpha = e, \mu, \tau$, $i = 1, 2, 3$ and $\mathcal{U}_{\alpha i}$ are the elements of the lepton-mixing Pontecorvo-Maki-Nakagawa-Sakata (PMNS) matrix [4, 5].

A neutrino created as one flavor state may be detected sometime later as another with probability

$$P_{\alpha \to \beta} = |\langle \nu_\beta | \nu_\alpha(t) \rangle|^2 = |\sum_i \mathcal{U}^*_{\alpha i} \mathcal{U}_{\beta i} e^{-iE_i t}|^2.$$

The dynamics of the neutrinos is governed by the Schrödinger equation

$$i\frac{\partial}{\partial t}|\nu\rangle = H_\nu|\nu\rangle, \tag{1}$$

where $H_\nu = diag\{E_1, E_2, E_3\}$ is the Hamiltonian of neutrinos in the mass eigenstates basis with $E_i$ representing the energy of $|\nu_i\rangle$. It could also be described by the Liouville–von Neumann equation using the density operator $\rho_\nu(t) = \sum_{ij} \rho_{ij}(t)|\nu_i\rangle\langle\nu_j|$:

$$\dot{\rho}_\nu(t) = -i[H_\nu, \rho_\nu(t)]. \tag{2}$$

Thus the evolution is unitary, and coherence (represented by the off-diagonal terms of the density matrix) is maintained during the propagation.

However, in general, we do see the loss of coherence. For example, solar neutrinos [6] are described as a mixture of incoherent mass eigenstates. There are several processes that may lead to the decoherence phenomenon (vanishing of the off-diagonal terms).

Wave packet dissipation [7–13] is one possible origin of decoherence, which is produced via the separation of neutrino mass states over long distances due to their different group velocities; this is still a unitary evolution and can be described as usual quantum mechanical framework described by Eqs. (1-2).

The other is environment-induced decoherence [14], which happens when the neutrinos are (weakly) coupled to the environment, becoming entangled with the environment as they propagate. The evolution is again unitary if we enlarge our Hilbert space to include the environment. However, due to the huge size of the environment and our ignorance, we have to trace out the environment's degrees of freedom, which leads to the emergence of non-unitarity.

Without knowing much of the details of the environment, master equations are useful tools to describe the time evolution of the neutrino density matrix $\rho_\nu$. For Markovian environments, which do not have a memory of their previous states, the most general type of master equation that is trace-preserving and completely positive is the Gorini–Kossakowski–Sudarshan–Lindblad (GKSL) equation [15, 16]:

$$\dot{\rho}_\nu(t) = -i[H_\nu, \rho_\nu] + \mathcal{L}_D[\rho_\nu], \tag{3}$$

where the first term is the same as the Liouville-von Neumann equation (2) representing the unitary dynamics, and $\mathcal{L}_D$ is the Lindblad decohering term, which characterizes the non-unitary decohering processes,

$$\mathcal{L}_D[\rho_\nu] = \sum_m (L_m \rho_\nu L_m^\dagger - \frac{L_m^\dagger L_m \rho_\nu + \rho_\nu L_m^\dagger L_m}{2}), \tag{4}$$

where the $L_m$ are the so-called Lindblad operators describing the influence of the environment on the system implicitly. There have been detailed studies on neutrino decoherence using the GKSL master equation formalism [17–31], and constraints on parameters in the decohering term $\mathcal{L}_D$ have been analyzed using experimental data in [32–37]. However, there is a lack of a general theoretical approach to deriving the Lindblad operators in terms of the neutrino interaction with the environment.

In this paper, we focus on the environment-induced decoherence and study neutrinos in an open quantum system. We first work out the evolution of neutrinos using the density matrix formalism. Then we look at two simple solvable models where the environment is described by forced harmonic oscillators [38] or two-level systems [39]. Next, we switch to the quantum field theory framework and calculate the decoherence rate due to scatterings with environmental particles. We show the universal exponential decay of coherence for weak couplings and Markovian processes, consistent with the GKSL master equation solutions. We then discuss the close connection between the Lindblad operators and the interaction Hamiltonians, in a way that gives a clear physical meaning of the Lindblad operators. Finally, we discuss extensions of the models to describe real physical processes with a more involved neutrino-environment interaction.

## II. DENSITY MATRIX FORMALISM

Consider neutrinos in interaction with the environment; the most general Hamiltonian reads

$$H = H_\nu \otimes I_\mathcal{E} + I_\nu \otimes H_\mathcal{E} + H_{\nu\mathcal{E}},$$

where $H_\nu = \sum_i E_i |\nu_i\rangle\langle\nu_i|$ describes the neutrino energy, $H_\mathcal{E}$ describes the environment $\mathcal{E}$, and $H_{\nu\mathcal{E}}$ is the Hamiltonian for the neutrino-environment interaction.

We focus on the case that $[H_\nu, H_{\nu\mathcal{E}}] = 0$, which implies energy conservation for the neutrino subsystem. The interaction term is then block diagonal and can be expressed by

$$H_{\nu\mathcal{E}} = \sum_i |\nu_i\rangle\langle\nu_i| \otimes H_{\mathcal{E}i}. \tag{5}$$

Assuming the neutrino is not entangled with the environment when created at $t = 0$, the initial density matrix describing the neutrino plus environment can be written as

$$\rho_0 = \rho_\nu \otimes \rho_\mathcal{E}.$$

The time development of this density matrix can be described by the evolution operator $U(t) = e^{-iHt}$:

$$\begin{aligned}
\rho(t) &= U(t)\rho_0 U^\dagger(t) \\
&= e^{-i(H_\nu + H_\mathcal{E} + H_{\nu\mathcal{E}})t}\rho_\nu \otimes \rho_\mathcal{E} e^{i(H_\nu + H_\mathcal{E} + H_{\nu\mathcal{E}})t} \\
&= \sum_{ij} \rho_{ij} e^{-iE_i t}|\nu_i\rangle\langle\nu_j|e^{iE_j t} \otimes e^{-iH_i t}\rho_\mathcal{E} e^{iH_j t},
\end{aligned} \tag{6}$$

where $H_i = H_\mathcal{E} + H_{\mathcal{E}i}$, $\rho_{ij} = \langle\nu_i|\rho_\nu|\nu_j\rangle$, with $i$ and $j$ labeling neutrino mass eigenstates.

For simplicity, we first look at two flavors of neutrinos, later generalizing the result for the case of three neutrinos. If an electron neutrino $|\nu_e\rangle = \cos\theta|\nu_1\rangle + \sin\theta|\nu_2\rangle$ is created at $t = 0$, we have

$$\rho_\nu = \rho_e = |\nu_e\rangle\langle\nu_e| = \begin{pmatrix} \cos^2\theta & \cos\theta\sin\theta \\ \cos\theta\sin\theta & \sin^2\theta \end{pmatrix}.$$

Taking the partial trace of Eq. (6) by summing over the environmental degrees of freedom, we get

$$\rho_\nu(t) = Tr_\mathcal{E}[\rho(t)] = \sum_{ij} Tr(e^{iH_j t}e^{-iH_i t}\rho_\mathcal{E})\rho_{ij}e^{-i\delta E_{ij}t}|\nu_i\rangle\langle\nu_j|,$$

where $\delta E_{ij} = E_i - E_j \approx \frac{\delta m_{ij}^2}{2E}$.

Define the decoherence form factor

$$F(t) = Tr(e^{iH_2 t}e^{-iH_1 t}\rho_\mathcal{E}) = \beta(t)e^{i\alpha(t)}, \tag{7}$$

where $\alpha(t)$ and $\beta(t)$ are real functions of time. We then have

$$\rho_\nu(t) = \begin{pmatrix} \cos^2\theta & F(t)\cos\theta\sin\theta e^{i\frac{\delta m_{ij}^2}{2E}t} \\ F^*(t)\cos\theta\sin\theta e^{-i\frac{\delta m_{ij}^2}{2E}t} & \sin^2\theta \end{pmatrix}.$$

The survival probability of the electron neutrino is [40]

$$P_{ee}(t) = Tr[\rho_e \rho_\nu(t)] = 1 - \frac{1}{2}\sin^2 2\theta [1 - \beta(t)\cos(\frac{\delta m_{ij}^2}{2E}t + \alpha(t))]. \qquad (8)$$

Let us look at some limiting cases of the form factor $F(t)$:

- When the environment is decoupled, we simply have $F(t) = 1$. In this case, neutrinos evolve unitarily and can be described by a phase rotation:

$$\rho(t) = R_z(\phi)\rho(0)R_z^\dagger(\phi),$$

where $\phi = \frac{\delta m_{ij}^2}{2E}t$ and $R_z(\phi)$ is the rotational operator around the z-axis of the Bloch sphere

$$R_z(\phi) = e^{i\phi\frac{\sigma_z}{2}} = \begin{pmatrix} e^{i\frac{\phi}{2}} & 0 \\ 0 & e^{-i\frac{\phi}{2}} \end{pmatrix}.$$

The survival probability of electron neutrino takes the standard form

$$P_{ee}(t) = 1 - \sin^2 2\theta \sin^2 \frac{\delta m_{ij}^2}{4E}t. \qquad (9)$$

- If $|F(t)| = 1$ with $\alpha(t) \neq 0$, the evolution of neutrinos is still unitary and can be described by the same rotational operator as the previous case. However, the rotational angle is altered to be

$$\phi(t) = \frac{\delta m_{ij}^2}{2E}t + \alpha(t),$$

through the influence of the environment. The survival probability then becomes

$$P_{ee}(t) = 1 - \sin^2 2\theta \sin^2 \left( \frac{\delta m_{ij}^2}{4E}t + \frac{\alpha(t)}{2} \right). \qquad (10)$$

Coherence stays as long as there exists a definite phase relation between $|\nu_1\rangle$ and $|\nu_2\rangle$. However, decoherence occurs when the inserted phase $\alpha(t)$ acts as random noise. If $\alpha$ is drawn from a Gaussian distribution with zero mean and variance $\gamma$:

$$p(\alpha) = \frac{1}{\sqrt{2\pi\gamma}}\exp[-\frac{\alpha^2}{2\gamma}],$$

the form factor after taking the ensemble average is given by

$$\langle F \rangle = \int_{-\infty}^{+\infty} F(\alpha)p(\alpha)d\alpha = \int_{-\infty}^{+\infty} \frac{1}{\sqrt{2\pi\gamma}}e^{i\alpha - \frac{\alpha^2}{2\gamma}}d\alpha = e^{-\frac{\gamma}{2}}.$$

The off-diagonal elements of the density matrix exponentially decay for increasing $\gamma$, which corresponds to the phase damping channel of decoherence for which we will look at an example in section IV.

- When $|F(t)| \to 0$ decoherence occurs due to the decay of the amplitude $\beta(t)$. We will look at an example of this case in the next section.

## III. ENVIRONMENT AS FORCED HARMONIC OSCILLATORS

In this section, we study a specific model which gives an explicit form of $F(t)$ and illustrate the decoherence process.

We model the environment by $N$ identical harmonic oscillators with Hamiltonian $H_{\mathcal{E}} = \sum_{s=1}^{N} \omega a_s^{\dagger} a_s$, where $a_s^{\dagger}$ and $a_s$ denote the creation and annihilation operators of the $s$'th harmonic oscillator correspondingly. The interaction with neutrinos is modeled as a linear coupling

$$H_{\nu\mathcal{E}} = (\lambda_1 |\nu_1\rangle\langle\nu_1| + \lambda_2 |\nu_2\rangle\langle\nu_2|) \otimes \sum_{s=1}^{N} f_s(t)(a_s^{\dagger} + a_s), \qquad (11)$$

which acts as random driving forces on those harmonic oscillators. This model has an exact analytic solution, which helps provide an aid in constructing approximations for more complicated systems.

For simplicity, assume that the environment is initially at the ground state (zero temperature):

$$\rho_{\mathcal{E}} = |0_{\mathcal{E}}\rangle\langle0_{\mathcal{E}}| = |0_1\rangle\langle0_1| \otimes |0_2\rangle\langle0_2| \otimes \cdots \otimes |0_N\rangle\langle0_N|.$$

Then the Hamiltonian $H_{is} = \omega a_s^{\dagger} a_s + \lambda_i f_s(t)(a_s^{\dagger} + a_s)$ evolves the $s$'th harmonic oscillator from its ground state to a coherent state

$$e^{-iH_{is}t}|0\rangle_s = e^{i\lambda_i^2 \eta_s(t)}|\lambda_i \zeta_s(t)e^{-i\omega t}\rangle, \qquad (12)$$

where

$$\zeta_s(t) = -i \int_0^t f_s(t')e^{i\omega t'} \, \mathrm{d}t', \qquad (13)$$

and

$$\eta_s(t) = \int_0^t \mathrm{d}t' \int_0^{t'} \mathrm{d}t'' f_s(t')f_s(t'') \sin(\omega(t' - t'')). \qquad (14)$$

Coherent states $|z\rangle$ are defined by

$$|z\rangle = e^{za_s^{\dagger} - z^* a_s}|0\rangle = e^{-\frac{|z|^2}{2}} e^{za_s^{\dagger}}|0\rangle = e^{-\frac{|z|^2}{2}} \sum_n \frac{z^n}{\sqrt{n!}}|n\rangle,$$

with

$$\langle z_1 | z_2 \rangle = e^{-\frac{1}{2}(|z_1|^2 + |z_2|^2 - 2z_1^* z_2)},$$

and with $z$ replaced by $\lambda_i \zeta_s(t) e^{-i\omega t}$ in Eq. (12).

Now we calculate the form factor

$$
\begin{aligned}
F(t) &\equiv Tr(e^{iH_2 t} e^{-iH_1 t} \rho_\mathcal{E}) \\
&= \prod_s \langle 0_s | e^{iH_{2s} t} e^{-iH_{1s} t} | 0_s \rangle \\
&= \prod_s e^{i(\lambda_1^2 - \lambda_2^2)\eta_s(t)} \langle \lambda_2 \zeta_s(t) e^{-i\omega t} | \lambda_1 \zeta_s(t) e^{-i\omega t} \rangle \\
&= \prod_s e^{-\frac{1}{2}(\lambda_1^2 + \lambda_2^2 - 2\lambda_1 \lambda_2)|\zeta_s(t)|^2 + i(\lambda_1^2 - \lambda_2^2)\eta_s(t)} \equiv \beta(t) e^{i\alpha(t)}.
\end{aligned}
$$

Define the autocorrelation function of the random variable $f_s(t)$:

$$
C_f(\tau) \equiv \langle f(t) f(t+\tau) \rangle = \frac{1}{N} \sum_s f_s(t) f_s(t+\tau),
$$

where we assumed that the random process $f_s(t)$ is stationary and ergodic, in which case time averages are equal to ensemble averages.

Further calculation gives

$$
\begin{aligned}
\beta(t) &= \exp[-\frac{1}{2}(\lambda_1^2 + \lambda_2^2 - 2\lambda_1\lambda_2) \sum_s |\zeta_s(t)|^2] \\
&= \exp[-\frac{(\lambda_1 - \lambda_2)^2}{2} \int_0^t dt' \int_0^t dt'' N C_f(t'' - t') e^{i\omega(t' - t'')}], \\
\alpha(t) &= (\lambda_1^2 - \lambda_2^2) \sum_s \eta_s(t) \\
&= (\lambda_1^2 - \lambda_2^2) \int_0^t dt' \int_0^{t'} dt'' N C_f(t'' - t') \sin(\omega(t' - t'')).
\end{aligned}
$$

Without knowing explicit details of $f(t)$, we look at a "white noise" power spectrum with the correlation function given by $C_f(\tau) = g\delta(\tau)$, which may correspond to the effects of vacuum fluctuations of the background field. In this case, the result simplifies to

$$
\alpha(t) = 0,
$$

$$
\beta(t) = \exp\{-\frac{1}{2}(\lambda_1 - \lambda_2)^2 Ngt\}.
$$

In the van Hove weak coupling limit, $N \to \infty$, $g \to 0$, but $gN \to G$ fixed, we obtain a simple exponential decay of the form factor

$$
F(t) = \exp\{-\frac{1}{2}(\lambda_1 - \lambda_2)^2 Gt\}, \tag{15}
$$

with the decoherence rate given by

$$
\Gamma = \frac{1}{2}(\lambda_1 - \lambda_2)^2 G.
$$

Note that if $\lambda_1 = \lambda_2 = \lambda$, there is no decoherence effect and we simply have the decoherence rate $\Gamma = 0$. In this case the neutrino part of the interaction $H_{\nu\mathcal{E}} = \lambda I \otimes \sum_{s=1}^{N} f_s(t)(a_s^\dagger + a_s)$ is proportional to the identity operator, which can not distinguish different species of neutrinos.

Generalizing to the case of three neutrinos, the Hamiltonian describing the neutrino-environment interaction reads

$$H_{\nu\mathcal{E}} = \sum_{i=1}^{3} \lambda_i |\nu_i\rangle\langle\nu_i| \otimes \sum_{s=1}^{N} f_s(t)(a_s^\dagger + a_s).$$

Following the same procedure of this section, we get three decoherence factors

$$F_{ij}(t) = \exp\{-\frac{1}{2}(\lambda_i - \lambda_j)^2 Gt\} = e^{-\Gamma_{ij}t}. \tag{16}$$

Again there is no decoherence effect when $\lambda_1 = \lambda_2 = \lambda_3$. When two of the coupling strengths are the same but the third one is different $\lambda_1 = \lambda_2 \neq \lambda_3$, coherence is maintained between $|\nu_1\rangle$ and $|\nu_1\rangle$, but $|\nu_3\rangle$ decoheres with them at the rate $\Gamma_{13} = \Gamma_{23}$. For three different coupling constants, we only need two independent parameters to describe the decoherence among three flavors of neutrinos. E.g. when $\lambda_1 < \lambda_2 < \lambda_3$, we have the relation

$$\sqrt{\Gamma_{13}} = \sqrt{\Gamma_{12}} + \sqrt{\Gamma_{23}}.$$

Now we look at a different limiting case for time independent driving forces $f_s(t) = f$. Plugging into Eqs. (13-14) we get

$$\zeta_s(t) = -\frac{2if}{\omega} e^{\frac{i\omega t}{2}} \sin\frac{\omega t}{2},$$
$$\eta_s(t) = \frac{f^2}{\omega^2}(\omega t - \sin\omega t).$$

In the van Hove limit $Nf^2 \to G$, we obtain

$$\alpha(t) = (\lambda_1^2 - \lambda_2^2)\frac{G}{\omega^2}(\omega t - \sin\omega t), \tag{17}$$

$$\beta(t) = \exp[-(\lambda_1 - \lambda_2)^2\frac{2G}{\omega^2}\sin^2\frac{\omega t}{2}]. \tag{18}$$

We observe that the form factor $|F(t)|$ is periodic with recurrence time $\omega^{-1}$, which means the loss of coherence is recoverable in this case. For times that are short compared to the environmental dynamics $t \ll \omega^{-1}$, the decay of coherence becomes Gaussian:

$$|F(t)| = \exp[-\frac{1}{2}(\lambda_1 - \lambda_2)^2 Gt^2]. \tag{19}$$

## IV. ENVIRONMENT AS TWO-LEVEL SYSTEMS

By the principle of universality, we do not expect our conclusions will be altered if we change our model of the environment. In this section we study a different model where the environment is described by a collection of $N$ two-level systems represented by $\{|\uparrow_s\rangle, |\downarrow_s\rangle\}$, $s = 1 \cdots N$. The free Hamiltonian of the environment is given by

$$H_\mathcal{E} = \sum_{s=1}^{N} \omega \sigma_z^s,$$

where $\sigma_z^s = |\uparrow_s\rangle\langle\uparrow_s| - |\downarrow_s\rangle\langle\downarrow_s|$.

We consider a bilinear neutrino-environment coupling that induces phase damping

$$H_{\nu\mathcal{E}} = (\lambda_1|\nu_1\rangle\langle\nu_1| + \lambda_2|\nu_2\rangle\langle\nu_2|) \otimes \sum_{s=1}^{N} f_s(t)\sigma_z^s.$$

For simplicity, we assume that the neutrino-environment system is initially factorizable:

$$|\Phi_{\nu\mathcal{E}}(0)\rangle = |\nu(0)\rangle \otimes |\mathcal{E}(0)\rangle = (\cos\theta|\nu_1\rangle + \sin\theta|\nu_2\rangle) \otimes \prod_{s=1}^{N}(a_s|\uparrow_s\rangle + b_s|\downarrow_s\rangle),$$

where $a_s$ and $b_s$ are random numbers satisfying $|a_s|^2 + |b_s|^2 = 1$.

Since the interaction $H_{\nu\mathcal{E}}$ commutes with both $H_\nu$ and $H_\mathcal{E}$, the evolution of the whole system can be directly written down as

$$
\begin{aligned}
|\Phi_{\nu\mathcal{E}}(t)\rangle &= e^{-iH_{\nu\mathcal{E}}t}\left(e^{-iH_\nu t}|\nu(0)\rangle \otimes e^{-iH_\mathcal{E}t}|\mathcal{E}(0)\rangle\right) \\
&= e^{-iE_1 t}\cos\theta|\nu_1\rangle \otimes |\mathcal{E}_1(t)\rangle + e^{-iE_1 t}\sin\theta|\nu_2\rangle \otimes |\mathcal{E}_2(t)\rangle,
\end{aligned}
$$

where

$$
\begin{aligned}
|\mathcal{E}_1(t)\rangle &= \exp[-i\sum_{s=1}^{N}(\omega t + \lambda_1\xi_s(t))\sigma_z^s]|\mathcal{E}(0)\rangle \\
&= \prod_{s=1}^{N}(e^{-i\omega t - i\lambda_1\xi_s(t)}a_s|\uparrow_s\rangle + e^{i\omega t + i\lambda_1\xi_s(t)}b_s|\downarrow_s\rangle), \\
|\mathcal{E}_2(t)\rangle &= \exp[-i\sum_{s=1}^{N}(\omega t + \lambda_2\xi_s(t))\sigma_z^s]|\mathcal{E}(0)\rangle \\
&= \prod_{s=1}^{N}(e^{-i\omega t - i\lambda_2\xi_s(t)}a_s|\uparrow_s\rangle + e^{i\omega t + i\lambda_2\xi_s(t)}b_s|\downarrow_s\rangle),
\end{aligned}
$$

with the time-dependent phase $\xi_s(t)$ defined by

$$\xi_s(t) = \int_0^t f_s(t)\mathrm{d}t. \tag{20}$$

We see that neutrinos are now entangled with the environment. The form factor is given by

$$F(t) = \langle \mathcal{E}_2(t)|\mathcal{E}_1(t)\rangle = \prod_{s=1}^{N}(e^{-i(\lambda_1-\lambda_2)\xi_s(t)}|a_s|^2 + e^{i(\lambda_1-\lambda_2)\xi_s(t)}|b_s|^2). \tag{21}$$

We model the neutrino-environment interaction as instantaneous kicks at discrete random times:

$$f_s(t) = \sum_r g_{s,r}\delta(t - t_r),$$

where $g_{s,r}$ is the kicking strength at time $t = t_r$, which is randomly drawn from a probability distribution $p(g_s)$.

The contribution to the form factor from one kicking event is

$$F_s(1) = e^{-i(\lambda_1-\lambda_2)g_{s,1}}|a_s|^2 + e^{i(\lambda_1-\lambda_2)g_{s,1}}|b_s|^2.$$

For $n$ successively applied kicks, we have

$$\begin{aligned}
F_s(n) &= e^{-i(\lambda_1-\lambda_2)\sum_{r=1}^n g_{s,r}}|a_s|^2 + e^{i(\lambda_1-\lambda_2)\sum_{r=1}^n g_{s,r}}|b_s|^2 \\
&= e^{-i(\lambda_1-\lambda_2)n\bar{g}_s}|a_s|^2 + e^{i(\lambda_1-\lambda_2)n\bar{g}_s}|b_s|^2.
\end{aligned}$$

The central limit theorem of probability states that, when a random process is driven by a large number of statistically independent, random influences, its probability becomes Gaussian:

$$p(\bar{g}_s) = \frac{1}{\sqrt{2\pi}\sigma_{\bar{g}_s}}\exp[-\frac{\bar{g}_s^2}{2\sigma_{\bar{g}_s}^2}],$$

where $\sigma_{\bar{g}_s} = \frac{\sigma_{g_s}}{\sqrt{n}}$, with $\sigma_{g_s}$ representing the standard deviation of the probability distribution $p(g_s)$. Therefore, after taking the ensemble average, we get

$$\langle F_s(n)\rangle = \int_{-\infty}^{+\infty} F_s(n)p(\bar{g}_s)\mathrm{d}\alpha = e^{-\frac{1}{2}(\lambda_1-\lambda_2)^2\sigma_{g_s}^2 n},$$

where we have used $|a_s|^2 + |b_s|^2 = 1$. Note that this result does not depend on the initial state.

For kicking events obeying a Poisson distribution with rate $\gamma_s$, where $\gamma_s$ denotes the average number of events per unit of time, and defining

$G_s = \gamma \sigma_{g_s}^2$ and replacing $n$ with $\gamma t$, we get the form factor as a function of time

$$F_s(t) = e^{-\frac{1}{2}(\lambda_1 - \lambda_2)^2 G_s t}.$$

With all environmental degrees of freedom included, defining $G = \sum_{s=1}^{N} G_s$, we obtain

$$F(t) = \prod_{s=1}^{N} F_s(t) = e^{-\frac{1}{2}(\lambda_1 - \lambda_2)^2 G t}. \tag{22}$$

As expected, this gives the same exponential decay as Eq. (15). The actual details of the environment become unimportant as a consequence of all our approximations. However, this study gives us a guide on how to go beyond these types of interactions.

For constant interaction terms $f_s(t) = f$, Eqs. (20-21) gives

$$F(t) = \prod_{s=1}^{N} (e^{-i(\lambda_1 - \lambda_2)ft}|a_s|^2 + e^{i(\lambda_1 - \lambda_2)ft}|b_s|^2),$$

so that

$$|F(t)|^2 = \prod_{s=1}^{N} \{1 - 4|a_s|^2|b_s|^2 \sin^2[(\lambda_1 - \lambda_2)ft]\}. \tag{23}$$

Again we see the periodic-like behavior of the form factor. As $t \to \infty$, we can take the time average $\langle \sin^2[(\lambda_1 - \lambda_2)ft] \rangle = \frac{1}{2}$ and obtain

$$\langle |F(t)|^2 \rangle \xrightarrow{t \to \infty} \prod_{s=1}^{N} \{1 - 2|a_s|^2|b_s|^2\} \xrightarrow{N \to \infty} 0. \tag{24}$$

Decoherence occurs as long as the environment contains a sufficient number of degrees of freedom.

## V.   A QUANTUM FIELD THEORETIC APPROACH

In this section, we calculate the decoherence rate of neutrinos due to their interactions with the environment in the quantum field theory framework. The interactions between the neutrinos and medium particles are described by the following effective Lagrangian

$$\mathcal{L} = (\lambda_1 \nu_1 \nu_1 + \lambda_2 \nu_2 \nu_2)\phi\phi, \tag{25}$$

where $\lambda_i$'s are the coupling constants. For simplicity, we ignore the spin of neutrinos and the medium particle $\phi$. The generalization to the case of an arbitrary spin of $\phi$ and spin-$\frac{1}{2}$ neutrinos is straightforward.

The initial state of an electron neutrino and a medium particle before scattering is assumed to be at their momentum eigenstates

$$|i\rangle = \mathcal{N}|\nu_e(\mathbf{p})\rangle|\phi(\mathbf{q})\rangle = \mathcal{N}(\cos\theta|\nu_1(\mathbf{p})\rangle + \sin\theta|\nu_2(\mathbf{p})\rangle)|\phi(\mathbf{q})\rangle$$
$$\equiv \mathcal{N}(\cos\theta|i_1\rangle + \sin\theta|i_2\rangle), \tag{26}$$

where $\mathcal{N}$ is a normalization factor. We adopt the Lorentz invariant normalization condition for the momentum eigenstates

$$\langle A(\mathbf{p'})|A(\mathbf{p})\rangle = 2E_A(\mathbf{p})(2\pi)^3\delta^{(3)}(\mathbf{p} - \mathbf{p'}), \tag{27}$$

where $|A(\mathbf{p})\rangle$ is the one particle momentum eigenstate corresponding to momentum $\mathbf{p}$ and energy $E_A(\mathbf{p}) = \sqrt{\mathbf{p}^2 + m_A^2}$, $m_A$ being the mass of the particle $A$.

In order to have $\langle i|i\rangle = 1$, we have

$$\mathcal{N}^{-1} = 2\sqrt{|\mathbf{p}|m_\phi}V, \tag{28}$$

where $V = (2\pi)^3\delta^{(3)}(0)$ is the normalization volume, and we have made the approximation $E_{\nu_1}(\mathbf{p}) \simeq E_{\nu_2}(\mathbf{p}) \simeq |\mathbf{p}|$ for relativistic neutrinos and $E_\phi(\mathbf{q}) \simeq m_\phi$ for the non-relativistic medium particle.

For an elastic scattering, the final state is given by $|f\rangle = S|i\rangle$, with $S = e^{-iHt} = 1 + i\mathcal{T}$. The S-matrix elements are given by

$$\langle \nu_i(\mathbf{p'})\phi(\mathbf{q'})|\mathcal{T}|\nu_i(\mathbf{p})\phi(\mathbf{q})\rangle = (2\pi)^4\delta^{(4)}(p + q - p' - q')\mathcal{M}_i(p, q, p', q'), \tag{29}$$

where $\mathcal{M}_i$ can be calculated with Feynman diagrams and $\mathcal{M}_i \sim \lambda_i$ to the first order approximation according to Eq. (25).

Inserting the identity operator

$$1 = \sum_X \int d\Pi_X |X\rangle\langle X|, \tag{30}$$

where

$$d\Pi_X = \prod_{j \in X} \frac{d^3p_j}{(2\pi)^3 2E_j(\mathbf{p})}, \tag{31}$$

the final state can be written as

$$|f\rangle = \mathcal{N}(\cos\theta|f_1\rangle + \sin\theta|f_2\rangle), \tag{32}$$

where

$$|f_1\rangle = |\nu_1(\mathbf{p})\phi(\mathbf{q})\rangle + i\int d\Pi_{\mathbf{p'},\mathbf{q'}}(2\pi)^4\delta^{(4)}(p + q - p' - q')\mathcal{M}_1(p, q, p', q')|\nu_1(\mathbf{p'})\phi(\mathbf{q'})\rangle, \tag{33}$$

$$|f_2\rangle = |\nu_2(\mathbf{p})\phi(\mathbf{q})\rangle + i\int d\Pi_{\mathbf{p'},\mathbf{q'}}(2\pi)^4\delta^{(4)}(p + q - p' - q')\mathcal{M}_2(p, q, p', q')|\nu_2(\mathbf{p'})\phi(\mathbf{q'})\rangle. \tag{34}$$

We see that the momentum eigenstates of neutrinos are entangled with the environment after scattering. The entanglement measures of $\nu_1$ and $\nu_2$ with environment are different as long as $\lambda_1 \neq \lambda_2$.

The initial and final density operator of the whole system can be written as

$$\rho_{\nu\phi,i} = |i\rangle\langle i|, \quad \rho_{\nu\phi,f} = |f\rangle\langle f|. \tag{35}$$

We can get the reduced density operator of neutrinos by tracing out the environmental degrees of freedom

$$\rho_\nu = Tr_\phi[\rho_{\nu\phi}] = \int \frac{V d^3 k}{(2\pi)^3} \frac{\langle \phi(\mathbf{k})|\rho_f|\phi(\mathbf{k})\rangle}{\langle \phi(\mathbf{k})|\phi(\mathbf{k})\rangle}. \tag{36}$$

The matrix element of $\rho_\nu$ is given by

$$\rho_{\nu,ij}(\mathbf{k}) = \frac{\langle \nu_i(\mathbf{k})|\rho_\nu|\nu_j(\mathbf{k})\rangle}{\sqrt{\langle \nu_i(\mathbf{k})|\nu_i(\mathbf{k})\rangle\langle \nu_j(\mathbf{k})|\nu_j(\mathbf{k})\rangle}}. \tag{37}$$

If all the outgoing neutrino states are measured, we would simply take an integral over the momentum variable and get a two-by-two matrix

$$\hat{\rho}_{\nu,ij} = \int \frac{V d^3 k}{(2\pi)^3} \rho_{\nu,ij}(\mathbf{k}). \tag{38}$$

The diagonal terms of $\rho_\nu$ stay unchanged due to unitarity $1 = S^\dagger S = (1 - i\mathcal{T}^\dagger)(1 + i\mathcal{T})$, e.g.

$$\hat{\rho}_{\nu,f,11} = \cos^2\theta = \hat{\rho}_{\nu,i,11}, \quad \hat{\rho}_{\nu,f,22} = \sin^2\theta = \hat{\rho}_{\nu,i,22}. \tag{39}$$

The off-diagonal term becomes

$$\hat{\rho}_{\nu,12} = \cos\theta\sin\theta(1 + F), \tag{40}$$

where the form factor is equal to

$$F = i\mathcal{N}^2 VT\mathcal{M}_1(p,q,p,q) - i\mathcal{N}^2 VT\mathcal{M}_2^*(p,q,p,q)$$
$$+ \mathcal{N}^2 VT \int d\Pi_{\mathbf{p}',\mathbf{q}'}(2\pi)^4\delta^{(4)}(p+q-p'-q')\mathcal{M}_1(p,q,p',q')\mathcal{M}_2^*(p,q,p',q'). \tag{41}$$

where $VT = (2\pi)^4\delta^{(4)}(0)$ and $\mathcal{N}$ is given by Eq. (28).

The decoherence rate is given by the real part of the form factor. Using the optical theorem, we have

$$\text{Im}[\mathcal{M}_i(p,q,p,q)] = 2|\mathbf{p}|m_\phi v\sigma_i, \tag{42}$$

where $v$ is the relative speed between the neutrino and environment, $\sigma_i$ is the total cross section of $\nu_i$ and $\phi$. The third term in (41) can be approximated as $\frac{Tv}{V}\sqrt{\sigma_1\sigma_2}$. So we have

$$\text{Re}[F] = -\frac{Tv}{2V}(\sigma_1 + \sigma_2) + \frac{Tv}{V}\sqrt{\sigma_1\sigma_2} = -\frac{Tv}{2V}(\sqrt{\sigma_1} - \sqrt{\sigma_2})^2. \qquad (43)$$

Consider the flux of medium with number density $N_\phi$ and integrate over time, we finally get

$$F(t) = e^{-\Gamma t + i\alpha(t)}, \qquad (44)$$

where $\Gamma = -\frac{N_\phi v}{2}(\sqrt{\sigma_1} - \sqrt{\sigma_2})^2$ is the decoherence rate, in agreement with Eq. (16) and Eq. (22) in the quantum mechanics formalism, which makes sense because the interaction Lagrangian (25) resembles the Hamiltonian (5) in the QM formalism.

The imaginary part $\alpha(t)$ of the form factor characterizes the index of refraction of the medium, and can be described by a correction to the Hamiltonian in the QM approach. The survival probability of electron neutrinos is then given by Eqs. (7-8).

A more consistent description of the decoherence process requires the wave packet approach in QM or a QFT treatment, where the wave packet depends on the production and detection process of the neutrinos. These aspects are out of the scope of this work and will be addressed in the future.

## VI. CONNECTIONS WITH THE GKSL MASTER EQUATION

Master equations are useful tools when little is known of the environment. The most general type of master equation for Markovian environments is the GKSL equation:

$$\dot{\rho}_\nu(t) = -i[H_\nu, \rho_\nu] + \mathcal{L}_D[\rho_\nu], \qquad (45)$$

where the first term describes the unitary evolution of the system just as the Liouville-von Neumann equation does, and the non-unitary decohering processes are characterized by the Lindblad decohering term $\mathcal{L}_D$:

$$\mathcal{L}_D[\rho_\nu] = \sum_m (L_m \rho_\nu L_m^\dagger - \frac{L_m^\dagger L_m \rho_\nu + \rho_\nu L_m^\dagger L_m}{2}), \qquad (46)$$

where the influence of the environment on the system is implicitly described by Lindblad operators $L_m$.

Several constraints can be made to reduce the number of parameters in the master equation. The Lindblad operators should be Hermitian $L_m =$

$L_m^\dagger$ to ensure that the von Neumann entropy $S = -Tr(\rho_\nu \ln \rho_\nu)$ increases in time [41]. To impose the energy conservation of neutrinos, the Lindblad operators should commute with the Hamiltonian $[H_\nu, L_m] = 0$. Consider the case where there is only one Lindblad operator of the following diagonal form:

$$L = diag\{l_1, l_2, l_3\}.$$

The solution of the master equation (45) is given by

$$\rho(t) = \begin{pmatrix} \rho_{11} & \rho_{12}e^{i\frac{m_{12}}{2E}t - \Gamma_{12}t} & \rho_{13}e^{i\frac{m_{13}}{2E}t - \Gamma_{13}t} \\ \rho_{21}e^{-i\frac{m_{12}}{2E}t - \Gamma_{12}t} & \rho_{22} & \rho_{23}e^{i\frac{m_{23}}{2E}t - \Gamma_{23}t} \\ \rho_{31}e^{-i\frac{m_{13}}{2E}t - \Gamma_{13}t} & \rho_{32}\rho_{23}e^{-i\frac{m_{23}}{2E}t - \Gamma_{23}t} & \rho_{33} \end{pmatrix},$$

where $\Gamma_{ij} = \frac{1}{2}(l_i - l_j)^2$. The solution exhibits the same exponential decay behavior as in Eq. (16) and Eq. (22) for $l_i = \lambda_i\sqrt{G}$, as expected by the principle of universality. In the QFT approach we have $l_i = \sqrt{N_\phi v \sigma_i}$, which is again propotional to $\lambda_i$. The essential features of the decoherence process are not affected by the particular details of the environment, as long as our model contains the basic ingredients to provide the effect. In the appendix, the Lindblad operator emerges naturally in the derivation of the GKSL master equation as a consequence of the Born-Markov approximation. The physical meaning of the Lindblad operators $\{L_i\}$ turn out to be the neutrino part of the interaction $H_{\nu\mathcal{E}}$, which means

$$H_{\nu\mathcal{E}} \propto L \otimes \sum_{s=1}^{N} f_s(t)(a_s^\dagger + a_s)$$

for the harmonic oscillator model, and

$$H_{\nu\mathcal{E}} \propto L \otimes \sum_{s=1}^{N} f_s(t)\sigma_z^s$$

for two-level systems.

When multiple types of interaction exist, there would be more than one Lindblad operator, and we have to do the summation as in Eq. (46), where each Lindblad operator corresponds to a particular type of interaction with the environment.

## VII. CONCLUSION AND OUTLOOK

This paper first studied two toy models of neutrinos in interaction with the environment with exact analytical solutions, which correspondingly illustrated two possible decoherence channels: phase damping and amplitude

damping. For weakly-coupled Markovian processes, the two distinct models give the same exponential decay of coherence

$$F_{ij}(t) = e^{-\Gamma_{ij}t}$$

where $\Gamma_{ij} \propto (\lambda_i - \lambda_j)^2$ and $\lambda_i$ is the coupling strength of the $i$'th species of neutrino that couples to the environment. The universality of the decoherence process is properly characterized by the Lindblad operator $L \propto diag\{\lambda_1, \lambda_2, \lambda_3\}$ and the result is consistent with the GKSL master equation.

A more consistent and accurate description of the decoherence effect requires a quantum field theoretic treatment. We then looked at a simple example of the QFT approach, which gives the same exponential dependence as the QM approach. The decay rate is expressed by $\Gamma_{ij} = -\frac{N_\phi v}{2}(\sqrt{\sigma_i} - \sqrt{\sigma_j})^2$, with the Lindblad operators related to the cross sections. A more generalized calculation with neutrinos and environment treated as wave packets in the QM or QFT framework will be addressed in the future.

Since the Born-Markov approximation is a fundamental condition for the emergence of the Lindblad operators, the GKSL master equation (45) is insufficient to describe non-Markovian processes such as Eq. (18) and Eq. (23). In such cases, we need to have a complete description of the environmental dynamics. This work on explicit models of the environment serves as a guide on how to alter the equation as a result of a more involved neutrino-environment interaction. With some modifications, the models we considered in this paper are suitable to describe real physical processes. For example, the environment as two-level systems could describe the scattering events with electrons and muons as neutrinos propagate through matter, where the decoherence effect will lead to a modification of the MSW mechanism [42, 43]. Moreover, to effectively model the decoherence from quantum gravity effects [44], one can extend the harmonic oscillator model to describe the environment as an ensemble of D-branes [45, 46] in thermodynamic equilibrium at Planck's temperature. These could be the direction of future works.

### ACKNOWLEDGMENTS

Bin Xu thanks Prof. Pierre Ramond, Prof. M. Jay Pérez, Prof. Alexander Stuart, and Moinul Hossain Rahat for helpful discussions and comments on the manuscript. This work was partially supported by the U.S. Department of Energy under grant number DE-SC0010296.

**Appendix A: A derivation of the GKSL master equation**

Here we present a derivation of the GKSL master equation for the forced harmonic oscillator model and show the relationship between the Lindblad operator and neutrino-environment interaction Hamiltonian.

Firstly, it is convenient to convert to the interaction picture of $H_\nu + H_\mathcal{E}$ with the following transformations:

$$\tilde{H}_{\nu\mathcal{E}} = U_0^\dagger(t) H_{\nu\mathcal{E}} U_0(t),$$
$$\tilde{\rho}(t) = U_0^\dagger(t) \rho(t) U_0(t),$$

where $U_0(t) = e^{-i(H_\nu + H_\mathcal{E})t}$.

Initially, the neutrinos are not entangled with the environment

$$\tilde{\rho}(0) = \rho(0) = \rho_\nu(0) \otimes \rho_\mathcal{E}(0),$$

The Hamiltonian in the interaction picture can be explicitly calculated as

$$\tilde{H}_{\nu\mathcal{E}} = \sum_i \lambda_i |\nu_i\rangle\langle\nu_i| \otimes \sum_s f_s(t)(a_s^\dagger e^{i\omega t} + a_s e^{-i\omega t}).$$

The equation for $\tilde{\rho}(t)$ in the interaction picture reads

$$i\dot{\tilde{\rho}}(t) = [\tilde{H}_{\nu\mathcal{E}}, \tilde{\rho}(t)], \tag{A.1}$$

which is equivalent to the integro-differential equation

$$\tilde{\rho}(t) = \tilde{\rho}(0) - i \int_0^t dt' [\tilde{H}_{\nu\mathcal{E}}(t'), \tilde{\rho}(t')]. \tag{A.2}$$

Inserting Eq. (A.2) into the right-hand side of Eq. (A.1) we obtain

$$\dot{\tilde{\rho}}(t) = -i[\tilde{H}_{\nu\mathcal{E}}, \tilde{\rho}(0)] - \int_0^t dt' [\tilde{H}_{\nu\mathcal{E}}(t), [\tilde{H}_{\nu\mathcal{E}}(t'), \tilde{\rho}(t')]].$$

The evolution for $\tilde{\rho}_\nu(t)$ is obtained by taking the partial trace of the environment's degrees of freedom

$$\dot{\tilde{\rho}}_\nu(t) = -i Tr_\mathcal{E}([\tilde{H}_{\nu\mathcal{E}}, \tilde{\rho}(0)]) - \int_0^t dt' Tr_\mathcal{E}([\tilde{H}_{\nu\mathcal{E}}(t), [\tilde{H}_{\nu\mathcal{E}}(t'), \tilde{\rho}(t')]]). \tag{A.3}$$

Eq. (A.3) is exact without any approximations. The first term on the right hand side turns out to be zero with $\rho_\mathcal{E}(0) = |0_\mathcal{E}\rangle\langle 0_\mathcal{E}|$:

$$Tr_\mathcal{E}([\tilde{H}_{\nu\mathcal{E}}, \tilde{\rho}(0)]) = \left[\sum_i \lambda_i |\nu_i\rangle\langle\nu_i|, \rho_\nu(0)\right] \cdot \langle 0_\mathcal{E}| \sum_s f_s(t)(a_s^\dagger + a_s)|0_\mathcal{E}\rangle = 0.$$

The GKSL equation appears naturally as a consequence of the Born and Markov approximations. For the Born approximation, we assume that the coupling is so weak, and the reservoir is so large that its state is unaffected by the interaction. Thus we can write

$$\tilde{\rho}(t) = \tilde{\rho}_\nu(t) \otimes \rho_\mathcal{E},$$

so that Eq. (A.3) becomes

$$\dot{\tilde{\rho}}_\nu(t) = -\int_0^t \mathrm{d}t' Tr_\mathcal{E}([\tilde{H}_{\nu\mathcal{E}}(t), [\tilde{H}_{\nu\mathcal{E}}(t'), \tilde{\rho}_\nu(t')\rho_\mathcal{E}]]).$$

For the Markov approximation, we replace $\tilde{\rho}_\nu(t')$ with $\tilde{\rho}_\nu(t)$ and obtain

$$\dot{\tilde{\rho}}_\nu(t) = -\int_0^t \mathrm{d}t' Tr_\mathcal{E}([\tilde{H}_{\nu\mathcal{E}}(t), [\tilde{H}_{\nu\mathcal{E}}(t'), \tilde{\rho}_\nu(t)\rho_\mathcal{E}]]). \tag{A.4}$$

Note that this equation is no longer integro-differential but simply differential, which means the density matrix evolves under a time-local first-order differential equation.

We now expand $H_{\nu\mathcal{E}}$ over the following form

$$H_{\nu\mathcal{E}} = \sum_m \sigma_m B_m,$$

where $\{\sigma_m\}$ is a basis of Hermitian operators acting on the neutrinos, which could be Pauli matrices or Gell-Mann matrices for two or three flavors of neutrinos correspondingly, and the operators $B_i$ act only on the environment. Converting to the interaction picture, we have

$$\tilde{H}_{\nu\mathcal{E}}(t) = \sum_m \tilde{\sigma}_m(t)\tilde{B}_m(t), \tag{A.5}$$

where $\tilde{\sigma}_m(t) = e^{iH_\nu t}\sigma_m e^{-iH_\nu t}$ and $\tilde{B}_m(t) = e^{iH_\mathcal{E}t}B_m e^{-iH_\mathcal{E}t}$. Comparing to Eq. (A), for the case of two neutrinos we have

$$\tilde{\sigma}_0(t) = I, \quad \tilde{\sigma}_3(t) = \sigma_3,$$

$$\tilde{B}_0(t) = \frac{\lambda_1 + \lambda_2}{2} \sum_s f_s(t)(a_s^\dagger e^{i\omega t} + a_s e^{-i\omega t}),$$

$$\tilde{B}_3(t) = \frac{\lambda_1 - \lambda_2}{2} \sum_s f_s(t)(a_s^\dagger e^{i\omega t} + a_s e^{-i\omega t}).$$

Plugging Eq. (A.5) into Eq. (A.4) and expanding the commutators we obtain

$$\dot{\tilde{\rho}}_\nu(t) = -\sum_{mn} \int_0^t \mathrm{d}t' [\tilde{\sigma}_m \tilde{\sigma}_n \tilde{\rho}_\nu(t)\Gamma_{mn}(t,t') - \tilde{\sigma}_m \tilde{\rho}_\nu(t)\tilde{\sigma}_n \Gamma_{nm}(t',t)$$
$$- \tilde{\sigma}_n \tilde{\rho}_\nu(t)\tilde{\sigma}_m \Gamma_{mn}(t,t') + \tilde{\rho}_\nu(t)\tilde{\sigma}_n \tilde{\sigma}_m \Gamma_{nm}(t',t)], \tag{A.6}$$

where

$$\Gamma_{mn}(t, t') = Tr_{\mathcal{E}}(\tilde{B}_m(t)\tilde{B}_n(t')\rho_{\mathcal{E}})$$

are the environment's correlation functions. For a memoryless white noise spectrum $\langle f(t)f(t+\tau)\rangle = g\delta(\tau)$, we simply have

$$\Gamma_{mn}(t, t+\tau) = \gamma_{mn}\delta(\tau), \qquad (A.7)$$

with $\gamma_{mn} = \gamma_{nm}$ and

$$\gamma_{00} = \frac{(\lambda_1 + \lambda_2)^2}{4}G, \gamma_{03} = \gamma_{30} = \frac{\lambda_1^2 - \lambda_2^2}{4}G, \gamma_{33} = \frac{(\lambda_1 - \lambda_2)^2}{4}G. \quad (A.8)$$

Replacing Eq. (A.7) in Eq. (A.6) we obtain

$$\dot{\tilde{\rho}}_\nu(t) = -\sum_{mn}\gamma_{mn}\{\tilde{\sigma}_m\tilde{\sigma}_n\tilde{\rho}_\nu(t) - \tilde{\sigma}_n\tilde{\rho}_\nu(t)\tilde{\sigma}_m - \tilde{\sigma}_n\tilde{\rho}_\nu(t)\tilde{\sigma}_m + \tilde{\rho}_\nu(t)\tilde{\sigma}_m\tilde{\sigma}_n\}.$$

Finally, returning to the Schrödinger picture we get

$$\dot{\rho}_\nu(t) = -i[H_\nu, \rho_\nu] + \sum_{mn}\gamma_{mn}\{[\sigma_n, \rho\sigma_m] + [\sigma_n\rho, \sigma_m]\}.$$

We can always diagonalize $\gamma_{mn}$ with a unitary matrix

$$\hat{\gamma} = S\gamma S^\dagger.$$

Let $\{\hat{\gamma}_m\}$ denote the eigenvalues of $\gamma$ and define $L_m = \sum_n \sqrt{2\hat{\gamma}_m}S_{mn}\sigma_n$; the GKSL equation becomes

$$\dot{\rho}_\nu(t) = -i[H_\nu, \rho_\nu] + \sum_m(L_m\rho_\nu L_m^\dagger - \frac{L_m^\dagger L_m\rho_\nu + \rho_\nu L_m^\dagger L_m}{2}).$$

There turns out to be only one Lindblad operator corresponding to Eq. (A.8)

$$L = \sqrt{\frac{G}{2}}(\lambda_1 + \lambda_2)\sigma_0 + \sqrt{\frac{G}{2}}(\lambda_1 - \lambda_2)\sigma_3 = \sqrt{2G}(\lambda_1|\nu_1\rangle\langle\nu_1| + \lambda_2|\nu_2\rangle\langle\nu_2|).$$

Comparing to Eq. (11), we see that the Lindblad operators $\{L\}$ is just the neutrino part of the interaction $H_{\nu\mathcal{E}}$:

$$H_{\nu\mathcal{E}} = \frac{L}{\sqrt{G}} \otimes \sum_s f_s(t)(a_s^\dagger + a_s).$$

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
