# Peer review of "Neutrino Decoherence in Simple Open Quantum Systems"

_SciPost Physics Core_

## Round 2 · Referee Report · Anonymous (Referee 1) · 2021-9-16

Strengths

The manuscript systematically studies the effects of interaction of certain forms with medium on the coherence of a relativistic quantum system.

Weaknesses

Although the study may be of interest from an academic point of view but the assumptions made on the interaction form do not hold valid for neutrino interaction with medium. Neutrino interaction with matter fields are either neutral current or charged current, neither of which satisfies the condition in Eq. (5) of the manuscript.
A neutral current interaction of mass (energy) eigenstate $\nu_i$ on the electron converts it into $\nu_e$ which is a coherent linear combination of all three mass eigenstates.
On the other hand, a charged current interaction converts it into a charged lepton. This effect is especially important for ultra high energy cosmic neutrinos with energies higher than $10^{18}$ eV traversing the Earth which have been studied in the literature.

Report

The manuscript in this present form is misleading because it misrepresents neutrino interaction form.

Requested changes

Realistic interaction form between neutrino and matter fields, as given by the standard electroweak interaction formalism, has to be considered.

In addition to the references listed in the manuscript, the papers with the following eprint numbers also study the problem of decoherence which should be added:
0803.0495; hep-ph/9802387; 2104.05806; 2005.03022; 1201.4128
2105.03272

---

## Editorial Decision

awaiting_resubmission